# PeerJ

# Regulation of the heat stress response in *Arabidopsis* by MPK6-targeted phosphorylation of the heat stress factor HsfA2

Alexandre Evrard[1,6], Mukesh Kumar[2,6], David Lecourieux[3,6], Jessica Lucks[4], Pascal von Koskull-Döring[5] and Heribert Hirt[1]

[1] URGV Plant Genomics Laboratory, Evry, France
[2] Department of Plant Molecular Biology, Max F. Perutz Laboratories, University of Vienna, Vienna, Austria
[3] UMR Ecophysiologie et Génomique Fonctionnelle de la Vigne, France
[4] Pharmazentrum frankfurt/ZAFES, Institute of Clinical Pharmacology, Goethe-University, Frankfurt, Germany
[5] Institute of Molecular Biosciences, Biocenter N200/R306, Goethe University, Frankfurt, Germany
[6] These authors contributed equally to this work.

## ABSTRACT

So far little is known on the functional role of phosphorylation in the heat stress response of plants. Here we present evidence that heat stress activates the Arabidopsis mitogen-activated protein kinase MPK6. *In vitro* and *in vivo* evidence is provided that MPK6 specifically targets the major heat stress transcription factor HsfA2. Activation of MPK6 results in complex formation with HsfA2. MPK6 phosphorylates HsfA2 on T249 and changes its intracellular localisation. Protein kinase and phosphatase inhibitor studies indicate that HsfA2 protein stability is regulated in a phosphorylation-dependent manner, but this mechanism is independent of MPK6. Overall, our data show that heat stress-induced targeting of HsfA2 by MPK6 participates in the complex regulatory mechanism how plants respond to heat stress.

Corresponding author
Heribert Hirt,
heribert.hirt@univie.ac.at

Subjects Plant Science
Keywords MAP kinase, Heat stress proteins, Heat stress transcription factors, Kinases, Arabidopsis

## INTRODUCTION

Due to their sessile lifestyle, plants are constantly subjected to a variety of biotic and abiotic stresses in their environment. To survive under these conditions, complex signalling networks developed which allow terrestrial plants to efficiently sense and respond to stressful conditions. In particular, temperatures above the optimum are sensed as heat stress, which disturbs cellular protein homeostasis. Accumulation of heat shock proteins (Hsps) are an essential part of the heat stress response and are assumed to play a central role in acquired thermotolerance in plants and other organisms (*Baniwal et al., 2004*; *Hartl & Hayer-Hartl, 2002*; *Haslbeck, 2002*; *Kotak, Vierling & Baumlein, 2007b*; *Morimoto, 1998*; *Nakamoto & Vigh, 2007*; *Wang et al., 2004*). The central regulators of the expression of heat

stress-responsive genes are the heat stress transcription factors (Hsfs) (*Baniwal et al., 2004*; *Kotak, Larkindale & Lee, 2007a*; *Miller & Mittler, 2006*; *Morimoto, 1998*; *Nover & Scharf, 1997*; *Scharf, Hohfeld & Nover, 1998*; *Schöffl, Prandl & Reindl, 1998*; *von Koskull-Doring, Scharf & Nover, 2007*).

It was shown that the transcriptional activity of mammalian Hsf1 is regulated by phosphorylation (*Knauf et al., 1996*; *Kline & Morimoto, 1997*; *Holmberg et al., 2001*). So far, 5 phosphorylation sites have been characterized for mammalian Hsf1: S230, S303, S307, S326 and S363 (*Chu et al., 1996*). During non-stress conditions Hsf1 is a monomer and phosphorylated at S303, S307, S326 and S363, which results in repression of DNA-binding capacity and transcriptional activity. During heat stress Hsf1 shuttles to the nucleus, assembles in heat stress granules, trimerizes and binds to heat stress elements. Ultimately, after phosphorylation of S230, it leads to transcriptional activation of Hsp genes (*Holmberg et al., 2001*; *Morimoto, 1998*; *Pirkkala, Nykanen & Sistonen, 2001*).

Another well investigated heat stress transcription system is that of yeast Hsf1 (ScHsf1) (*Sorger & Pelham, 1988*; *Sorger, 1991*). ScHsf1 is phosphorylated in response to heat shock and oxidative stress (*Sorger, Lewis & Pelham, 1987*; *Høj & Jakobsen, 1994*) while the phosphorylation state of this Hsf regulates both activation and inactivation of the transcription activation function (*Hashikawa & Sakurai, 2004*). A C-terminal modulator domain (CTM) is essential for activation of heat stress element-containing genes but also for hyperphosphorylation of the ScHsf1 during heat stress. The serine and threonine residues of the CTM are constitutively phosphorylated under normal conditions. After heat shock further phosphorylation is induced (*Sorger, 1990*).

In contrast to yeast and Drosophila with a single Hsf and mammals with up to four Hsfs, the plant Hsf family with 20–30 members is more complex (*Baniwal et al., 2004*; *von Koskull-Doring, Scharf & Nover, 2007*). The *Arabidopsis* Hsf family consists of 21 members belonging to three classes A, B and C (*Nover et al., 2001*; *Baniwal et al., 2004*; *von Koskull-Doring, Scharf & Nover, 2007*). From *Arabidopsis*, several Hsfs have been functionally characterized in more detail but so far nothing is known about their regulation by posttranslational modification (*Baniwal et al., 2007*; *Charng et al., 2007*; *Czarnecka-Verner et al., 2000*; *Czarnecka-Verner et al., 2004*; *Davletova et al., 2005*; *Hübel & Schöffl, 1994*; *Hübel, Lee & Schöffl, 1995*; *Kim & Schöffl, 2002*; *Kotak, Larkindale & Lee, 2007a*; *Li et al., 2005*; *Lohmann et al., 2004*; *Nishizawa et al., 2006*; *Panchuk, Volkov & Schoffl, 2002*; *Prändl et al., 1998*; *Reindl & Schöffl, 1998*; *Schramm et al., 2006*; *Wunderlich, Werr & Schoffl, 2003*).

Mitogen-activated protein kinases (MPKs) show a similar complexity of 20 genes in *Arabidopsis* (*Ichimura, 2002*). In a phylogenetic tree the conserved amino acid motif TxY, a phosphorylation site for MAPKKs, classifies the MPKs into two subtypes: a TEY subtype with a TEY motif and a TDY subtype with a TDY motif. MPKs with the TEY motif can be classified into subgroups A, B and C. Group D MAPKs are characterized by a TDY motif and an extended C-terminal region. Group A and B MPKs comprise evolutionary conserved common docking domains (CD-domains) in their C-terminal regions (*Ichimura, 2002*; *Tanoue et al., 2000*). This domain structure facilitates docking to

MAPKKs, but also protein phosphatases and substrates. Two adjacent amino acids (D and E) are crucial for interaction with a cluster of basic amino acids (K und R) of MAPKKs in the sequence ([LH][LHY]Dxx[DE]xx[DE]EPxC) conserved in this CD-domain (*Tanoue et al., 2000*). The group A members MPK3 and MPK6 are involved in various environmental stress and hormone responses (*Nühse et al., 2000*; *Ichimura et al., 2000*; *Desikan et al., 2001*; *Yuasa et al., 2001*). With the exception of MPK4 and MPK11, which are also involved in stress responses (*Ichimura et al., 2000*; *Desikan et al., 2001*; *Teige et al., 2004*; *Bethke et al., 2011*), little is yet known about the specific functions of most other group B and even less is known on group C or group D members, except that group C members (MPK1, 2, 7 and 14) and at least one group D member (MPK8) are downstream of MKK3 which plays a role in pathogen and oxidative stress homeostasis (*Döczi et al., 2007*; *Takahashi et al., 2011*).

So far, there are only few reports concerning an involvement of MAPKs as part of the heat stress response. A heat-induced MAPK activation was shown in tomato and alfalfa, but their molecular roles have not been elucidated further (*Link et al., 2002*; *Sangwan & Dhindsa, 2002*). With the experiments presented here we reveal a potential involvement of phosphorylation events as part of the heat stress response in *Arabidopsis* and unravel a molecular mechanism how MPK6 negatively regulates the heat stress response.

## MATERIALS AND METHODS

### Plant materials and growth conditions

The *Arabidopsis thaliana* ecotype Columbia (Col) was used as wild-type for developmental experiments as indicated. The *mpk6* mutant was described earlier (*Nakagami et al., 2006*). Plants were grown on 0.5X MS medium (Sigma) under long day condition (16 h light/8 h dark) with a humidity level of 50% and 50 μE cool white light. Stress treatments were performed according to *Charng et al. (2007)*.

### Transient expression assays

Transient expression assays were performed using *Arabidopsis* suspension cell culture and tobacco mesophyll protoplasts, respectively, as described (*Forreiter, Kirschner & Nover, 1997*; *Nakagami, Kiegerl & Hirt, 2004*; *Lyck et al., 1997*; *Scharf et al., 1998*) with slight modifications. Protoplasts were isolated and transformed by PEG (polyethylene glycol) mediated transformation at room temperature (25 °C) under dark conditions. Plant expression vectors used are based on the pRT series of vectors (*Töpfer, Schell & Steinbiss, 1988*; *Döring et al., 2000*). The expression vector for 3HA-HsfA2 and the reporter construct $P_{Hsp18-CI::}GUS$ was described by *Schramm et al. (2006)*.

### Inhibitors

To investigate the effect of phosphorylation and dephosphorylation on heat stress response, several broad-spectrum inhibitors were tested. All inhibitors were provided from Calbiochem, dissolved in DMSO as recommended and used at the following final concentrations: Staurosporin (Cat No. 569396) 10 μM, Cantharidin (Cat. No. 210155) 10 μM, Calyculin (Cat. No. 208851) 0.25 μM and Okadaic acid (Cat. No. 459618) 2 μM.
## In-gel kinase assays

C-terminal domains (CTD) of several Hsfs were cloned into the pDEST-15 vector (Invitrogen) and expressed in *E. coli* strain Rosetta (Novagen). Sequences of CTD-Hsfs proteins are described in Supplemental Table S1. GST-tagged Hsf proteins were purified using Glutathione sepharose 4B resin (Amersham) according the manufacturer recommendations. Cell extracts were prepared at different times after heat stress in extraction buffer (25 mM Tris $\pm$ HCl pH 7.8, 15 mM EGTA, 75 mM NaCl, 1 mM dithiothreitol (DTT), 10 mM $MgCl_2$, 1 mM NaF, 0.5 mM $NaVO_3$, 15 mM $\beta$-glycerophosphate, 15 mM 4-nitrophenylphosphate, 0.1% Tween-20, 0.5 mM phenylmethylsulfonyl fluoride (PMSF), 5 mg/ml leupeptin, 5 mg/ml aprotinin). After centrifugation at 20000xg for 45 min, the cleared supernatant was used. For in-gel protein kinase reactions, cell extracts containing 20 µg of total protein per lane were separated by SDS-PAGE. Myelin basic protein (MBP; 0.5 mg/ml) was used as a substrate polymerized in the polyacrylamide gel. After protein renaturation, the kinase reactions were performed in the gel as described (*Usami et al., 1995*). The gels were dried and analyzed by autoradiography.

## Immunocomplex kinase assays

Protoplast extracts containing equal protein amounts (100 µg) were subjected to a 2-h preincubation in the presence of 20 µl of mixed protein A- and G-Sepharose beads (1:1). The supernatant was then immunoprecipitated with 5 µl of anti-HA monoclonal antibody and 20 µl of protein G-Sepharose beads overnight at 4 °C. The beads were washed three times with wash buffer (50 mM Tris, pH 7.4, 250 mM NaCl, 5 mM EGTA, 5 mM EDTA, 0.1% Tween 20) and once with kinase buffer (20 mM HEPES, pH 7.4, 10 mM $MgCl_2$, 5 mM EGTA, and 1 mM dithiothreitol). Kinase reactions of the immunoprecipitated proteins were performed in 15 µl of kinase buffer containing 5 µg of MBP, 0.1 mM ATP, and 2 µCi of ($\gamma$-$^{32}$ P) ATP. The protein kinase reactions were performed at room temperature for 30 min, and the reactions were stopped by adding 4x SDS loading buffer. The phosphorylation of MBP was analyzed by autoradiography after separation on 15% SDS-PAGE.

## Immunoblotting

Immunoblotting was performed with equal amounts of protein separated by 10% SDS-PAGE and transferred to PVDF membranes (Millipore) by electroblotting. Membranes were probed with A-purified M23 (raised against MPK6), or with either anti-HA monoclonal antibody or with anti-Myc polyclonal antibody (A-14, Santa Cruz Biotechnology) or with anti-GFP antibody (sigma). Membranes were developed by enhanced chemiluminescence as recommended by the manufacturer (Gene Image, Amersham Biosciences).

## Confocal imaging using *Arabidopsis* mesophyll protoplasts

HsfA2 WT and mutated sequences were PCR amplified using the pRTHsfA2 vector (*Döring et al., 2000*) as a template. Primers used for PCRs are listed in Supplemental Table S2. For site directed mutagenesis a first PCR using the following pairs were generated:

Hsfa2NS-GW2-CterF/Hsfa2-249A-R (PCR1); Hsfa2NS-GW2-CterF/Hsfa2-T249D-R (PCR2); Hsfa2NS-GW2-R/Hsfa2-T249A-F (PCR3) and Hsfa2NS-GW2-R/Hsfa2-T249D-F (PCR4). PCR1 and PCR3 or PCR2 and PCR4 products were diluted and mixed together and then re-amplified with the Hsfa2NS-GW2-CterF/Hsfa2NS-GW2-R primer pairs to generate HsfA2-T249A and HsfA2-T249D mutants. pRTHsfA2 was also amplified with Hsfa2NS-GW2-CterF/Hsfa2NS-GW2-R to generate the HsfA2-WT version. The three PCR products were then introduced in pDNR207 (Invitrogen) trough a classical Gateway-BP reaction. After sequence verification HsfA2 constructs were introduced in p2GWF7 (*Karimi, Depicker & Hilson, 2007*) by a Gateway-LR reaction (Invitrogen) to give HsfA2-WT-GFP, HsfA2-T249A-GFP, and HsfA2-T249D-GFP plasmids. MPK7 construct was amplified with MPK7NS-GW2-CterF/MPK7NS-GW2-R to generate MPK7-GFP plasmid and was used as a control for immunoblotting (see below). Transient expression assays were performed using *Arabidopsis* mesophyll protoplasts isolated and transformed as described (*Yoo, Cho & Sheen, 2007*). Confocal imaging were performed with a Leica TCS-SP2 confocal microscope and analysed with the Leica confocal software (LCS). Autofluorescence and overlay pictures are described in Supplemental Figure S1.

## Mass spectrometry

Coomassie blue stained phosphorylated and non-phosphorylated HsfA2 protein bands were excised from the SDS-PAGE. Ion trap mass spectrometry was performed as according to the user manual and as described in *de la Fuente van Bentem et al. (2006)*.

## RESULTS AND DISCUSSION

### MPK6 is activated by heat stress

To get insights into the protein kinases regulated in response to heat stress, we performed in-gel kinase assays of protein extracts from *Arabidopsis* leaves before and after exposure to heat stress (Fig. 1). Detached leaves were incubated in pre-heated (37 °C) liquid medium and then placed at 37 °C for 1 h. As shown by in-gel kinase assays, several MBP-phosphorylating protein kinases in the range between 40–72 kDa become activated by heat stress (Fig. 1A). The strongest activity corresponds to a protein kinase with an approximative molecular mass of 48 kDa that displayed a maximum activation within 2 min of heat stress before returning to lower levels by 20 min. The transient activation of the 48 kDa protein kinase that is able to phosphorylate MBP in Arabidopsis resembled to the class of mitogen-activated protein kinases that was previously reported in tomato and alfafa (*Link et al., 2002*; *Sangwan & Dhindsa, 2002*). We confirmed this hypothesis by immunocomplex kinase assays with a specific antibody against Arabidopsis MPK6 (Fig. 1B). The corresponding immunoblotting experiments indicated that the observed heat-induced kinase activity was not due to changes in MPK6 protein amounts (Fig. 1B). These results indicate that in Arabidopsis, MPK6 might be the major heat stress transducer in plant cells and the activation of this MAPK is a highly conserved process in plants. These results confirm also the HS induced MPK6 activity detected in Arabidopsis leaves obtain by *Li et al. (2005)*.

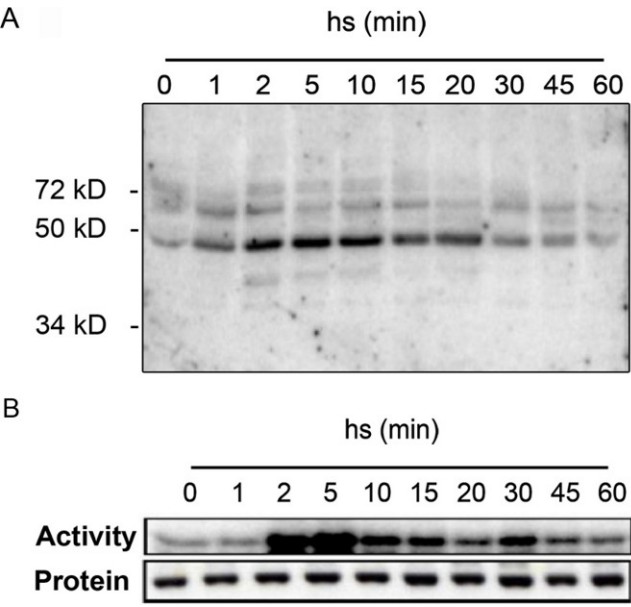

**Figure 1 Heat stress-induced MPK6 activation in *Arabidopsis* leaves.** After pre-incubation in liquid medium at 24 °C for 2 h, detached Arabidopsis leaves were transferred to pre-heated (37 °C) liquid medium and then placed at 37 °C for up to 60 min (hs: heat stress). Total protein extracts were used to perform in-gel protein kinase assays (A), and immunocomplex kinase assays with protein gel blot analysis (B). For the in-gel protein kinase assays, 20 μg of total proteins per lane were separated by SDS-PAGE. MBP (0.5 mg/ml) was used as a substrate polymerized in the polyacrylamide gel. MPK6 activity was determined at different times by immunocomplex kinase assay using 5 μg of protein A-purified MPK6-specific peptide M23 antibody. To monitor the protein levels, equal aliquots of the same extracts were analyzed by protein gel blot analysis using αMPK6 antibodies.

With in-gel kinase assays we could observe in samples from heat-stressed leaves one dominant heat-activated protein kinase of approximately 48 kDa (Fig. 1A) similar to that reported by *Link et al. (2002)* for tomato. MBP phosphorylation has been also detected to a lower extent for proteins in the range around 60 kDa proteins (Fig. 1A) and these kinases might also participate in the heat stress response. However, by immunocomplex assays with specific antibodies, we identified MPK6 as the most prominent heat stress-induced protein kinase in whole cell extracts (Fig. 1B). These results show that heat stress activation of MAPKs is also a highly conserved process in plants and add another factor to the list of activating conditions of MPK6, which was so far reported to be activated upon cold, salt, oxidative, hypoosmotic, ozone and genotoxic stress (*Droillard et al., 2004*; *Lee & Ellis, 2007*; *Teige et al., 2004*; *Ulm et al., 2002*; *Yuasa et al., 2001*), plant defence to biotic stress (*Brader et al., 2007*; *Menke et al., 2004*; *Schweighofer et al., 2007*) as well as jasmonate and ethylene signalling (*Ouaked et al., 2003*; *Takahashi et al., 2007*; *Schweighofer et al., 2007*).

## Phosphorylation of heat stress transcription factors by MPK6

In order to address the possibility that MPK6 could affect the heat stress response by direct phosphorylation of Hsfs as the regulators of the heat stress response, we performed *in vitro* kinase assays using active MPK6 on 16 recombinant GST-Hsf fusion proteins, representing all Hsf protein subgroups. To clarify the specificity of the enzymatic interactions, the

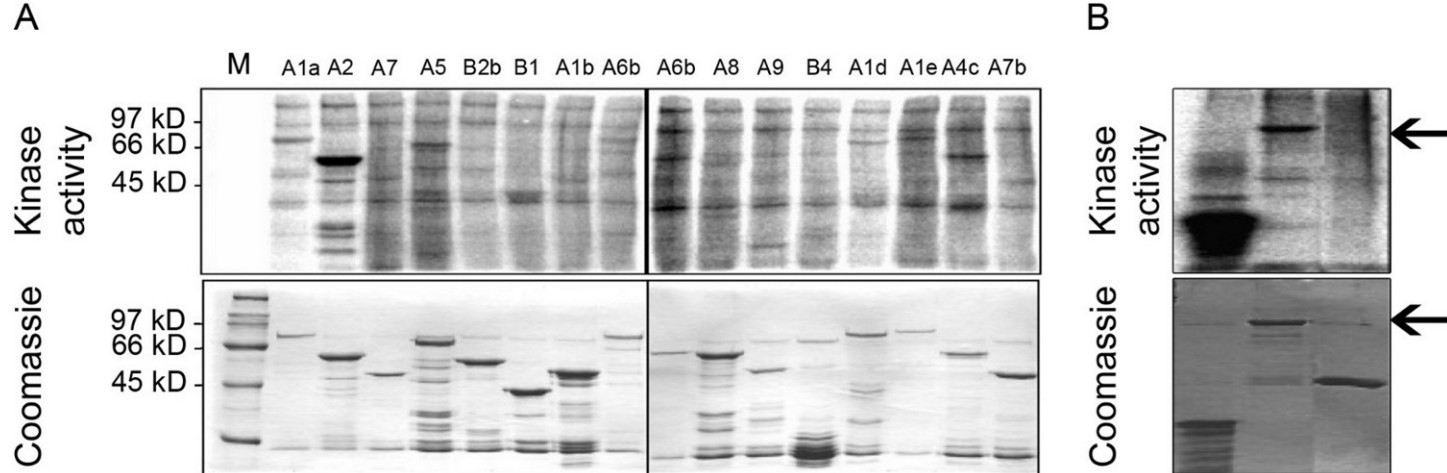

**Figure 2** *In vitro* **phosphorylation of heat stress transcription factors by MPK6.** (A) C-terminal domains (CTD) of several Hsfs were fused to GST. Using an HA-antibody, MPK6-HA was immunoprecipitated from Arabidopsis mesophyll protoplasts treated with 10 mM $H_2O_2$ for 10 min to activate MPK6 (*Rentel et al., 2004*). MPK6 was incubated with the purified GST-Hsfs for *in vitro* kinase assays with $^{32}P$-$\gamma$-ATP. The samples were separated on SDS-PAGE before Coomassie blue staining and autoradiography. M: molecular weight marker. (B) Similar assays as performed in (A) using in addition to GST-HsfA2 MBP and GST as controls. GST fusion proteins of HsfA2 is indicated by an arrow.

GST-Hsf fusions carried only the divergent C-terminal regulatory domains but not the highly conserved N-terminal DNA binding domains (see *Kotak et al., 2004*). As shown in Fig. 2A, MPK6 was able to phosphorylate several GST-Hsfs *in vitro*, but GST-HsfA2 appeared to be most strongly phosphorylated. In order to validate that the phosphorylation reactions towards the Hsf domains were specific, *in vitro* kinase assays were also performed with GST and MBP as negative and positive controls, respectively. Whereas MPK6 strongly phosphorylated MBP and GST-HsfA2, no phosphorylation was detected for GST (Fig. 2B).

Because transcription factors are common targets of MAPKs, we undertook a systematic *in vitro* screen for MPK6 targets among the 21 Arabidopsis Hsfs. Out of 16 candidates, representing all Arabidopsis Hsf subfamilies, MPK6 phosphorylated several members of different Hsf subclasses, but showed strong phosphorylation of HsfA2 (Fig. 2A). MPK6 was also shown to phosphorylate full length HsfA2 (Fig. 2B). Recently, putative targets of MPK3 and 6 have been described by *Sörensson et al. (2012)*. HsfA4e appear to contain the PPSPR motif and might be another target of MPK6.

### *In vivo* complex formation between MPK6 and HsfA2

The *in vitro* kinase assays (Fig. 2B) indicated that MPK6 is able to phosphorylate HsfA2. To corroborate that HsfA2 is a specific target of MPK6, we also tested whether MPK6 can directly interact with HsfA2 *in vivo*. For this purpose, we transiently expressed MPK6 and full-length wild type HsfA2 in *Arabidopsis* cells under the constitutive 35S Cauliflower Mosaic virus (CaMV) promoter (Fig. 3). MPK6 and HsfA2 were expressed as tagged proteins with either the Myc-epitopes or the Hemagglutinin-(HA), respectively, to allow their immunoprecipitation and immunodetection. Protoplasts co-expressing MPK6-Myc and 3HA-HsfA2 were then incubated at 37 °C for 5 min before protein extraction. Using an

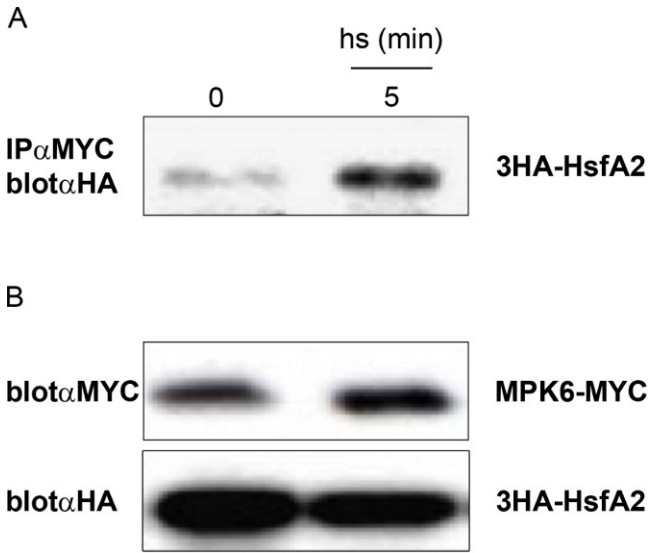

**Figure 3** *In vivo* **complex formation of MPK6 with HsfA2.** 3HA-HsfA2 was coexpressed with MPK6-Myc in *Arabidopsis* protoplasts. 16 h after transformation, protoplast samples were harvested before (0) or after incubation in a water bath at 37 °C for 5 min (5). (A) Total protein extracts were immunoprecipitated (IP) with αMyc antibody, and the immunoprecipitates were immunoblotted (Blot) for 3HA-HsfA2 with αHA antibody. (B) To monitor the protein levels, part of the same extracts were analyzed by protein gel blot analysis using αHA and αMyc antibodies.

anti-Myc antibody, MPK6-Myc was immunoprecipitated, and the presence of 3HA-HsfA2 in the complex was assayed by protein gel blot analysis using anti-HA antibody (Fig. 3A). HsfA2 was found to interact strongly with MPK6 at 5 min after heat stress treatment.

Although both 3HA-HsfA2 and MPK6-Myc were produced under the 35S CaMV promoter, we also tested whether the protein amounts changed during the course of the experiment. Anti-Myc immunoblotting showed that the protein amounts of MPK6-Myc and 3HA-HsfA2 did not change during the heat treatment (Fig. 3B). Overall, these experiments indicate that HsfA2 is a target of MPK6.

## MPK6 phosphorylates HsfA2 on threonine 249

To determine the MPK6 phosphorylation sites in HsfA2, we performed an *in vitro* kinase assay by incubating the recombinant GST-HsfA2 protein in the presence of cold ATP with or without active MPK6. The reaction products were separated by SDS-PAGE and excised after Coomassie blue staining. Upon in-gel alkylation-tryptic digestion, each digested sample was subjected to mass spectrometric analysis. No phosphorylated residue was detected in the nonphosphorylated sample, whereas a single phosphopeptide with the sequence 245RLTSTPSLGTMEENLLHDQEFDR268 was found for HsfA2 that was incubated with active MPK6 (Fig. 4A). A single phosphorylation site corresponded to the 247TST249 motif, but could not be identified to a specific residue within this sequence with certainty. We next mutated the 247TST249 in HsfA2 to 247AAA249 and expressed the mutant protein as recombinant GST fusion protein. When tested with active MPK6, the mutant HsfA2 247AAA249 protein could not be phosphorylated any more (unpublished

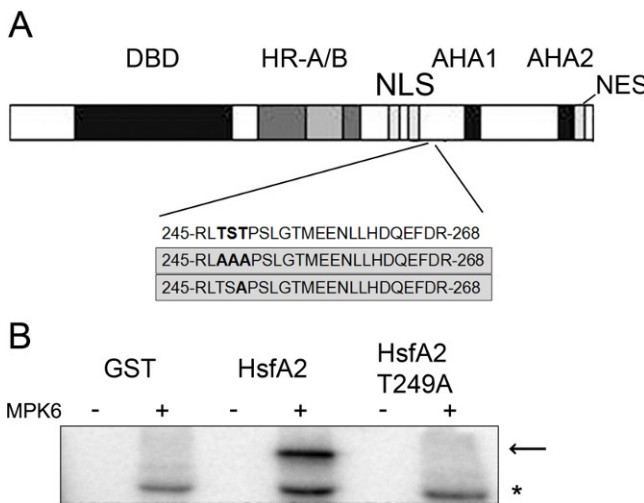

**Figure 4** **MPK6 phosphorylate HsfA2 on T249.** (A) Block Diagram of HsfA2 and the single phosphorylated T249 site identified by mass spectrometry. The functional domains involved in DNA-binding (DBD), oligomerization (HR-A/B), nuclear import (NLS) and nuclear export (NES) as well as its transcription activation potential (AHA) are indicated (*Nover et al., 2001*; *Schramm et al., 2006*). The mutated T249 motif is shown in the lower panel. (B) Mutation of T249 (HsfA2-T249A) abrogates phosphorylation of GST-HsfA2 by MPK6. GST fusion proteins of HsfA2 and HsfA2-T249A are indicated by an arrow, whereby GST as control is indicated by an asterisk.

results). Since MAP kinases absolutely require a proline residue in the +1 position following the serine/threonine phosphorylation site, we then only mutated T249 to alanine (T249A). As shown in Fig. 4B, MPK6 phosphorylated wild type GST-HsfA2 but not GST-HsfA2 T249A, indicating that T249 is the amino acid residue that is phosphorylated by MPK6. However, because of the presence of additional putative phosphorylation sites within the HsfA2 protein sequence we cannot rule out that other sites might be targeted by MPK6 or other kinases and play a role in its regulation.

## T249 phosphorylation regulates heat stress-induced nuclear accumulation of HsfA2

T249 lies in close proximity to the nuclear localization sequence (NLS) of HsfA2 (Fig. 4A), suggesting that phosphorylation of T249 could affect the localization of HsfA2. To analyse the possible consequences of phosphorylation of T249, we mutated T249 not only to alanine T249A to disable phosphorylation but also to glutamic acid T249D to mimic a phosphorylated status. Wild type HsfA2 (HsfA2-WT) and mutant T249A (HsfA2-T249A) or T249D (HsfA2-T249D) were then fused with GFP and expressed in Col-0 protoplasts under the constitutive 35S CaMV promoter. At ambient room temperature of 22 °C (RT), wild type HsfA2-WT and mutant HsfA2-T249A and HsfA2-T249D proteins showed similar cytoplasmic and nuclear localization (Fig. 5A, Col-0, RT). Upon heat stress, wild type and HsfA2-T249D accumulated in the nuclear compartment, whereas HsfA2-T249A was still distributed between cytoplasm and nucleus as in untreated cells (Fig. 5A, Col-0, HS). Threonine 249 that lies close to the nuclear localization signal of HsfA2, suggests that phosphorylation of this site might affect the

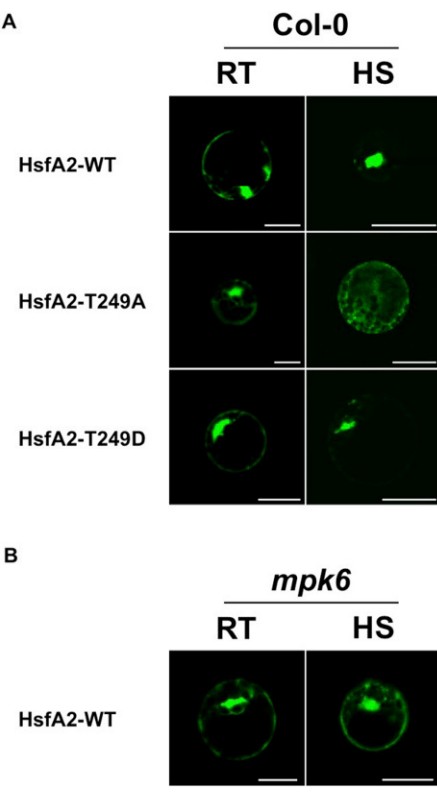

**Figure 5** **Analysis of the subcellular distribution of HsfA2 during a heat stress.** (A) HsfA2-WT-GFP, HsfA2-T249A-GFP, and HsfA2-T249D-GFP constructs were used to transform Col-0 Arabidopsis mesophyll protoplasts. Preceding confocal analysis, protoplasts were incubated for 4 h to 22 °C (RT) or 37 °C (HS). (B) HsfA2-WT-GFP construct was used to transform *mpk6* Arabidopsis mesophyll protoplasts and same experimental conditions were applied as in (A). Images are confocal scanning laser micrographs of protoplasts showing specific GFP fluorescence. Excitation wavelength was 495–530 nm for GFP and 636–676 nm for autofluorescence. Scale bar is 20 μm. Subcellular localization analysis was carried out at least three times with similar results.

intracellular localization of HsfA2. Localization studies performed in this work confirmed this hypothesis proving that T249 phosphorylation regulates the nuclear localization of HsfA2 upon heat stress. However we cannot discriminate whether nuclear localization of HsfA2 upon heat stress is a result of a cytosolic instability or a basic shuttling between the cytosol and the nucleus.

## MPK6 regulates heat stress-induced nuclear accumulation of HsfA2

To clarify whether MPK6 is responsible for the heat stress-induced nuclear accumulation, we then tested the behaviour of the HsfA2-GFP construct in protoplasts derived from *mpk6* knock out plants (Fig. 5B). Therefore, we expressed the wild type HsfA2-GFP construct in protoplasts prepared from *mpk6* knock out plants at ambient room temperature of 22 °C (RT) or 37 °C (HS). Under ambient conditions or heat stress conditions, HsfA2-GFP showed a similar distribution between nucleus and cytoplasm (Fig. 5B). Under heat stress conditions, HsfA2-GFP showed nuclear accumulation only in Col-0 derivative protoplasts,

indicating that MPK6 is the responsible factor that induces nuclear shuttling of HsfA2. S320 Phosphorylation of the human Hsf1 by a protein kinase plays a significant role in it nuclear localization (*Murshid et al., 2010*). Results presented in our work confirm that phosphorylation events are conserved mechanisms among organisms and kingdoms and regulate important biological responses to stress.

## Phosphorylation-dependent regulation of HsfA2 protein levels upon heat stress

Because HsfA2 phosphorylation can affect HsfA2 function we used protein kinase and phosphatase inhibitors in Arabidopsis mesophyll protoplasts. Upon application of the broad-spectrum protein kinase inhibitor staurosporine, or the protein phosphatase inhibitors cantharidin, calyculin and okadaic acid, the expression levels of endogenous HsfA2 or one of its targets, the heat stress-induced protein Hsp18.1-CI (*Schramm et al., 2006*) was determined in Arabidopsis cells (Fig. 6A). After application of the inhibitors, the cells were split into two aliquots and one aliquot was heat stressed for 3 h at 37 °C whereas the other aliquot was kept at 22 °C. Applying cantharidin had no marked effects on HsfA2 protein level in heat stressed cells (Fig. 6A). However, application of the kinase inhibitor staurosporin as well as the phosphatase inhibitor okadaic acid resulted in dramatically reduced levels of HsfA2 protein and consequently blocked almost completely the expression of sHsp-CI (Fig. 6A). These experiments indicated that phosphorylation and dephosphorylation events can influence the heat stress response of cells. Yet, it was not clear if HsfA2 and its downstream target Hsp18-CI were affected at the gene expression or protein levels, or if the observed effects were due to indirect effects of the inhibitors. Therefore, the inhibitors were also investigated in cells that constitutively express 3HA-HsfA2 under the 35S CaMV promoter but in the absence of heat stress. Protein extracts of the differently treated cells were probed with HA antibody in Western blots. DMSO as the solvent of the inhibitors alone had no marked effect on 3HA-HsfA2 levels, but both staurosporine as well as the three phosphatase inhibitors resulted in strongly reduced 3HA-HsfA2 protein levels and GUS accumulation mediated by the Hsp18.1-CI promoter (Figs. 6B and 6C). The changes in 3HA-HsfA2 protein levels were not due to a general toxic effect on the cells as protein amounts of Rubisco did not change significantly (Fig. 6B). These results show that phosphorylation-dependent mechanisms regulate HsfA2 protein amounts *in vivo*.

Using broad-spectrum protein kinase and phosphatase inhibitors resulted in a strong reduction in HsfA2 protein levels in Arabidopsis cells upon heat stress activation (Fig. 6). Since HsfA2 is regulated at both, the transcriptional and post-translational level (*Busch, Wunderlich & Schöffl, 2005*; *Schramm et al., 2008*; *Cohen-Peer et al., 2010*; *Ikeda, Mitsuda & Ohme-Takagi, 2011*), we also tried to uncouple these two levels of regulation by testing whether these effects are also observed when HsfA2 was permanently expressed under the constitutive 35S CaMV promoter. Under these conditions, we observed that inhibition of phosphorylation still reduced HsfA2 protein abundance upon heat stress except for

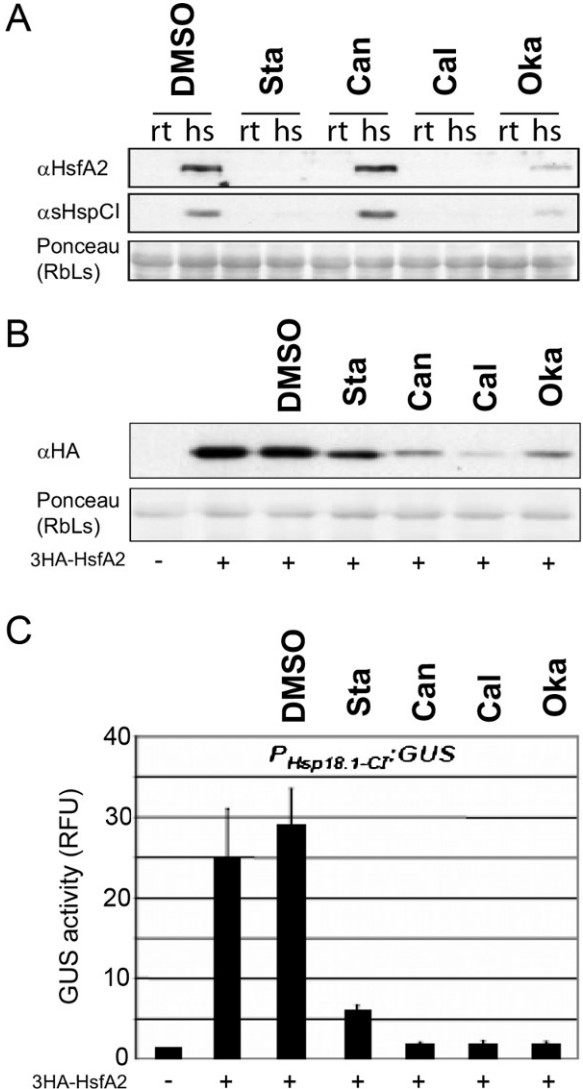

**Figure 6 Protein kinase and phosphatase inhibitors affect the expression of heat stress-induced proteins.** (A) *Arabidopsis* mesophyll protoplasts were treated with the inhibitors indicated (Sta: Staurosporin; Can: Cantharidin; Cal: Calyculin and Oka: okadaic acid) or mock treated with 1% DMSO (DMSO). After 30 min samples were split and kept either at RT as controls (rt) or heat stressed for three hours at 36 °C (hs). After another 60 min samples were harvested, proteins extracted and analyzed by immunoblot analysis with the antibodies indicated (*Schramm et al., 2006*). (B) The protein stability potential of 3HA-HsfA2 was tested in tobacco protoplasts in the presence of protein kinase and phosphatase inhibitors and (C) a reporter construct contained 1 kb of the Hsp18.1-CI promoter fused to GUS. The resulting GUS activities (relative fluorescence units, RFU) and protein stability are presented after co-transformation with the *35S::3HA-HsfA2* constructs (+) and inhibitors as in (A). Error bars correspond to the standard deviation of three independent replicates.

Cantharidin for which no marked effect has been observed in a native promoter condition (Fig. 6). These data suggest that these drugs might target different protein phosphatase complexes associated with plant heat stress signalling.

| | 1 | 2 | 3 | 4 | 5 |
|---|---|---|---|---|---|
| HsfA2 | - | + | + | + | + |
| MPK6 | - | - | + | + | + |
| MKK2 | - | - | - | + | - |
| MKK2$^{GOF}$ | - | - | - | - | + |
| αHA | | | | | |
| αMKK2 | | | | | |

**Figure 7 MPK6 does not regulate HsfA2 protein stability.** 3HA-HsfA2 protein abundance is evaluated by immunoblot analysis (2 to 5) from protoplast extracts co-expressing MPK6 (3 to 5) or an empty vector (1–2). Co-expression of active MKK2$^{GOF}$ is used to activate MPK6 in protoplasts. Co-expression of native form of MKK2 was used as control. MKK2 amounts are estimated by immunoblot analysis.

## Phosphorylation-dependent HsfA2 protein abundance regulation depends on other protein kinases than MPK6

Our work so far revealed that HsfA2 is a specific substrate of MPK6 (Fig. 2) that interacts with MPK6 in a heat stress-dependent manner (Fig. 3A). We also obtained evidence that HsfA2 protein abundance is regulated by phosphorylation (Fig. 5). To test this hypothesis, we performed Western blots of 3HA-HsfA2 upon co-expression of MPK6-Myc alone, or together with a constitutively active form of an MPK-activating MAPKK MKK2 (MKK2$^{GOF}$), in Arabidopsis cells under the constitutive 35S CaMV promoter (Fig. 7). Under these conditions, MPK6 is directly activated by MKK2$^{GOF}$ and no heat stress is necessary for its activation. To differentiate between activated and non-activated MPK6, HsfA2 and MPK6 were also co-expressed with the wild type form of MKK2. When HsfA2 was co-expressed with MPK6 or together with MKK2 or MKK2$^{GOF}$, HsfA2 levels were not affected when compared to expression of HsfA2 alone (Fig. 7). As a control, we also co-expressed HsfA2 with the MKK2 wild type and GOF version without MPK6. Under both conditions, HsfA2 protein levels did not show any reduction when compared to HsfA2 expression alone. To confirm that MPK6 was not involved in HsfA2 stability, we compared the level of HsfA2-GFP protein in Col-0 versus *mpk6* protoplasts under heat stress conditions (Supplemental Figure S2). Under these conditions, degradation of HsfA2 wasn't affected in the absence of MPK6. These data indicate that activated MPK6 does not regulate HsfA2 protein abundance. We conclude that other yet unidentified sites and most likely other protein kinases regulate HsfA2 protein stability.

## CONCLUSION

The heat stress response is a complex mechanism to protect plants against cell damage through protein degradation and aggregation during high temperatures. In mammals, the components of this molecular regulation machinery are well studied, while for plants only little is known. In yeast and mammals, phosphorylation was found to regulate the activity of Hsfs as part of the heat stress response (*Holmberg et al., 2002*; *Sorger & Pelham, 1988*; *Pirkkala, Nykanen & Sistonen, 2001*). One major class of kinases are the mitogen-activated protein kinases (MAPKs). In the stress response of *Arabidopsis* they were found to be involved in various signal transduction pathways, but up to now, a role of MAPKs in the Arabidopsis heat stress response was not defined. The present study demonstrates that MPK6 in Arabidopsis is activated by heat stress and targets the transcription factor

HsfA2. This work confirms also the results obtained by *Li et al. (2005)* where they nicely demonstrated that implication of one Vacuolar processing Enzyme ($\gamma$VPE) in HS induced cell death was MPK6 dependent. We also confirmed the *mpk6* tolerance to heat shock (Supplemental Figure S3). *Li et al. (2005)*, demonstrated that $\gamma$VPE expression and activity was dependent on MPK6 function, however, whether direct $\gamma$VPE phosphorylation might be associated with its activity remains to be investigated in the future. Also, the $\gamma$VPE encoding gene contains a highly conserved palindromic HSE in it promoter ((−605 *AGAA*AAAGA*TTCT* −593), (*Bienz & Pelham, 1987*; *Nover, 1991*; *Schramm et al., 2006*)). It would be interesting to investigate in further experiments whether HsfA2 could be a direct transcriptional regulator of the $\gamma$VPE gene.

Our data show that MPK6 interacts with HsfA2 *in vivo* upon heat stress. Moreover, we could show that MPK6 phosphorylates T249 of HsfA2 and thereby contributes to regulate the intracellular localization of HsfA2. Protein kinase and phosphatase studies indicated that HsfA2 protein stability is regulated in a phosphorylation-dependent manner. However, our studies showed that this mechanism is independent of MPK6. These data are compatible with the fact that expression, abundance and activity of HsfA2 are subject to multiple regulatory mechanisms. First, it was recently shown that Arabidopsis HsfB1 and HsfB2b negatively regulate the expression of several heat-inducible Hsfs, including HsfA2, and thereby several heat shock proteins (*Ikeda, Mitsuda & Ohme-Takagi, 2011*). In addition, HsfA1d and HsfA1e were identified to positively regulate HsfA2 gene expression by interacting with heat stress elements in the 5′-flanking region of the HsfA2 promoter (*Nishizawa-Yokoi et al., 2010*). It was also shown that the proteasome inhibitor MG132 affects expression of HsfA2 and its downstream targets (*Nishizawa-Yokoi et al., 2010*). Although the authors did not analyze whether MG132 also affects HsfA2 protein stability, there is evidence that the activity of the HsfA2 protein is regulated by post-translational mechanisms. *Cohen-Peer et al. (2010)* showed that SUMOylation of HsfA2 occurs *in vitro* and *in vivo* and regulates the potential of HsfA2 to activate downstream targets. Taken together, it is evident that HsfA2 is a key transcriptional regulator of the heat stress response that is subjected to multi-level regulation itself and that MPK6 only contributes to a certain degree to the full complexity of its regulation.

## ACKNOWLEDGEMENTS

We thank Gisela Englich for her excellent technical help.

### Funding

This work was supported by grants from the Austrian Science Foundation, the Deutsche Forschungsgemeinschaft (AFGN grant KO2888-1-1/2) and the French National Agency ANR. The funders had no role in study design, data collection and analysis, decision to publish, or preparation of the manuscript.

## Grant Disclosures

The following grant information was disclosed by the authors:

Austrian Science Foundation, the Deutsche Forschungsgemeinschaft.

AFGN: KO2888-1-1/2.

French National Agency ANR.

## Competing Interests

Heribert Hirt is an Academic Editor for PeerJ. There are no other competing interests.

## Author Contributions

- Alexandre Evrard and David Lecourieux conceived and designed the experiments, performed the experiments, analyzed the data, wrote the paper.
- Mukesh Kumar performed the experiments, analyzed the data, wrote the paper.
- Jessica Lucks performed the experiments, analyzed the data.
- Pascal von Koskull-Döring and Heribert Hirt conceived and designed the experiments, analyzed the data, wrote the paper.

## Data Deposition

The following information was supplied regarding the deposition of related data:

Sequence data from this article can be found in the GenBank/EMBL data libraries under accession numbers: MPK6 (At2g43790), HsfA2 (At2g26150).

## Supplemental Information

Supplemental information for this article can be found online at http://dx.doi.org/10.7717/peerj.59.

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
