# Peer review of "Regulation of the heat stress response in Arabidopsis by MPK6-targeted phosphorylation of the heat stress factor HsfA2"

_PeerJ, doi:10.7717/peerj.59_

## Round 0.1 · original submission · Major Revisions

The reviewers raised the important points to improve the quality of this manuscript. The reviewer 1 suggested some additional experiments, however, many of such results were already published in Li et al. (2012) New Phytol. 195: 85-96, which is missing in the references. The authors need to cite this paper as a key reference and explain the in vivo role of MPK6 in the heat stress response in Li et al. 2012. As the reviewer 2 commented, some results are novel. I recommend the authors to clearly point out the novel aspects of this work citing recent publications.

In addition, the following is my comments.
1) I suggest you to specify “MPK6” in the title rather than ambiguously stating “MAPK-targeted”, because this work does not characterize other MAPKs.

2) The sequence information for 16 HSFs used for the Fig. 2A experiment needs to be described in the M&M.

3) Does Fig 6 miss the pictures for T249A/mpk6 and T249D/mpk6?

4) some other editorial corrections: (i) page 6 M&M, ecotypes; not the plural, but is singular, (ii) page 6 M&M, Columbia (Col) should not be italic. (iii) page 7 M&M, E. coli should be italic (and C is not the large letter). (iv) page 7, MgCl2, NaVO3, the numbers should be the subscript. (v) page 7, 5 mg/ ml aprotinin, no space between / and ml. (vi) page 7, SDS+PAGE, SDS-PAGE, (vii) page 14 line 1, 35 S, no space between 35 and S. (viii) Figures can be formatted better, For example, Fig. 6C indicates #1~#7 at the bottom, but no explanations in the legend. I understand that each sample corresponds to that of Fig. 6B, but you need to make these figures and legends better. In addition, some figures show the control as “c” and the heat stress as “hs”, but another figure shows the control and heat stress as “RT” and “HS”. Please make these more consistent.

Reviewer 1 ·

Basic reporting

This manuscript reports that heat stress activates MPK6 leading to regulation of HsfA2 intracellular distribution for target gene induction via specific phosphorylation. Authors first showed that heat stress-induced activation of several protein kinases by in-gel kinase assay. Using MPK6 specific antibody, authors found MPK6 was included in heat-induced protein kinase. Heat-activated MPK6 increased affinity to bind HsfA2 and phosphorylated 249T of HsfA2. Heat-induced phosphorylation alters intracellular distribution of HsfA2. In addition to these result, analyses using inhibitors of protein kinases and phosphatases suggested that HsfA2 accumulation is regulated by phosphorylation. Connection however between this phenomena and MPK6 is not clear.
Manuscript preparation itself is premature since a lot of careless mistakes can be found, for example swapping fig 5 for fig 6 in the text, garbled symbol font characters, incorrect figure (figure 4 phosphorylation site sequences) and missing figure 7B, etc. The authors must thoroughly check whether they fixed bugs and errors before submission.

Experimental design

In the page 9 bottom, the authors write “we identified MPK6 as the most prominent heat stress-induced protein kinase in whole cell extract”. To say that, authors should check whether 48 kDa major band is MPK6. Although several bands can be seen in the figure 1A, only MPK6 has been studied in this manuscript. Considering MKK2 was used in figure7, kinase activity of MPK3 as well as MPK4 should be checked to see activation specificity among MPKs by heat stress. This is important point to say only MPK6 regulates HsfA2 or not. In figure 3, more MPKs should be used to reveal specific complex formation MPK6 and HsfA2. Subcellular localization of Wt and mutated form of HsfA2 was examined in the figure 5. It looks phosphorylation at T249 positively regulates cytoplasmic exclusion or HsfA2 instability in cytosol, not simple nuclear localization because GFP signal in the nucleus is observed before or after heat stress. In case of HsfA2-T249A, I don’t see clear nuclear localization after heat stress. The authors are able to show where HsfA2 and MPK6 interact before and after heat stress (when activation both proteins should bind at nucleus). The authors also studied role of protein phosphorylation and dephosphorylation on HsfA2 protein level and target gene induction. The author describe that regulation of HsfA2 accumulation is independent of MPK6. To say that, mpk6 mutant protoplast should be examined in figure 6A. In addition, it is hard to evaluate the result of figure 6B. Lastly, is there any evidence that heat-induced activation of MPK6 is MKK2 dependent?

Validity of the findings

I would like to see more solid data and interpretation about regulation of HsfA2 by heat-stress induced MPK6 and MPK specificity. In addition, it’s difficult to find out importance and connection of inhibitor study on HsfA2 accumulation and heat-induced activation of MPK6. Moreover, I would like to see thermo stress tolerance of mpk6 mutant and gain-of-function of pathway component to evaluate this manuscript.

Reviewer 2 ·

Basic reporting

This manuscript is well written and nicely prepared. MAP-kinases play multifaceted roles in development and stress signalling. Understanding how MAP-kinases are involved in these important, but diverse, biological responses requires the identification of their downstream substrates. In their manuscript Evrard et al. provide evidence that the heat stress transcription factor HsfA2 changes its intracellular localisation upon phosphorylation through MAP-kinase MPK6 in Arabidopsis thaliana. This is the first time that the phosphorylation of heat stress transcription factor by MPK6 and their direct interaction was demonstrated. In a recent paper (Biochem. J., 2012, 446: 271–278) describing the screening for MPK3 and MPK6 targets in Arabidopsis thaliana heat stress transcription factor A4a was identified as containing the putative MPK3/MPK6 consensus phosphorylation site P-P-S-P-R. Results presented in that publication should be included in the discussion of the present manuscript before acceptance for publication.

Experimental design

All the experiments are well designed and the results are clearly presented. The obtained results can be used to draw the conclusions presented in the manuscript.

Validity of the findings

The findings give novel insight into the specific roles of protein phosphorylation by MAP kinases in the stress response of plants. The discussion is sound and does not over interprete the results.

---

## Round 0.2 · Minor Revisions

Both reviewers and I agreed the revised manuscript is suitable for publication after minor editorial corrections.

Reviewer 1 ·

Basic reporting

Evrard et al., Regulation of the heat stress response in Arabidopsis by MPK6-targeted phosphorylation of the heat stress factor HsfA2
In the revised manuscript, the authors cited a key paper by Li et al. (2012), and discussed in the conclusion about possible in vivo role of MPK6 under heat stress condition. Together with the paper by Ki et al. (2012), the authors answered most questions raised in the first manuscript. They however did not fully respond to critical ones, it seems that the authors don’t have to do it.
The manuscript is now fully improved, but I still see some small points to be corrected:
page 5, line 15, “(25o C)” close space.
page 6, line 2, “0,25uM”
page 6, line 13, “20 000 g” should be 2000 xg?
page 7, line 3 “MgCl2” the number should be subscript.
page 7, line 5 “32P” the numbers should be superscript.

Experimental design

No Comments

Validity of the findings

No Comments

Reviewer 2 ·

Basic reporting

No comments

Experimental design

The experiments are well designed and the results are clearly presented in the figures.

Validity of the findings

The authors can draw the conclusion that the heat stress transcription factor HsfA2 directly interacts with MAP-kinase MPK6 in Arabidopsis thaliana and changes its intracellular localisation upon phosphorylation.

Additional comments

To the revised manuscript the requested additions were made. However, there are still mistakes that give the impression that the manuscript has not been revised with adequate care. The text format is not consistent and spelling mistakes were not corrected.
Page 6, line 8: resine should be resin.
Page 9, line 18 confirms should be confirm.
Page 17, line 10: confirm should be confirms. Line 13: Heat shock should be heat shock.
References:
Li C, et al. is not in the right format. Please erase the issue number and give the full last page number.
Murshid A et al. Please give the right format
Sörensson C et al. Please give the right format

---

## Round 0.3 · Minor Revisions

The revised manuscript has not been corrected the followings: (1) page 6 line 2, 0,25 to 0.25 (2) Sörensson C paper, correct Sep 1;446(2):271 to 446:271.

---

## Round 0.4 · accepted · Accept

The revised manuscript corrected all requested aspects. Thank you for submitting your work to PeerJ.